# MD3F: Multivariate Distance Drift Diffusion Framework for High-Dimensional Datasets

**DOI:** 10.3390/genes15050582

**Published:** 2024-05-03

**Authors:** Jessica Zielinski, Patricia Corby, Alexander V. Alekseyenko

**Affiliations:** 1Department of Oral Health Sciences, Medical University of South Carolina, Charleston, SC 29425, USA; zielinsk@musc.edu; 2Department of Radiation Oncology, University of Pittsburgh, Pittsburgh, PA 15260, USA; pat.corby@pitt.edu; 3Biomedical Informatics Center, Department of Public Health Sciences, College of Medicine, Medical University of South Carolina, Charleston, SC 29425, USA

**Keywords:** multivariate analysis, longitudinal omics, drift-diffusion model, microbiome

## Abstract

High-dimensional biomedical datasets have become easier to collect in the last two decades with the advent of multi-omic and single-cell experiments. These can generate over 1000 measurements per sample or per cell. More recently, focus has been drawn toward the need for longitudinal datasets, with the appreciation that important dynamic changes occur along transitions between health and disease. Analysis of longitudinal omics data comes with many challenges, including type I error inflation and corresponding loss in power when thousands of hypothesis tests are needed. Multivariate analysis can yield approaches with higher statistical power; however, multivariate methods for longitudinal data are currently limited. We propose a multivariate distance-based drift-diffusion framework (MD3F) to tackle the need for a multivariate approach to longitudinal, high-throughput datasets. We show that MD3F can result in surprisingly simple yet valid and powerful hypothesis testing and estimation approaches using generalized linear models. Through simulation and application studies, we show that MD3F is robust and can offer a broadly applicable method for assessing multivariate dynamics in omics data.

## 1. Introduction

Modern high-throughput assays, including transcriptomics, genomics, metagenomics, and metabolomics, provide a detailed lens into biologic systems and have accelerated biomarker discovery [1,2]. Longitudinal omics study designs are feasible in many settings now and allow for comparison before, after, and during treatments, across disease states or clinical conditions (drug resistance, etc.). For example, studies with a pre vs. post design have allowed for insights into the innate immune response to SARS-CoV2 in infants compared to adults, how gut dysbiosis and plasma metabolites contribute to amyotrophic lateral sclerosis, and the transcriptomic phenotypes that are associated with therapy-resistant multiple myelomas [3,4,5]. Longitudinal omics studies are further advantageous because they allow for the consideration of both within-subject variability and between-subject variability, as stated in Kodikara et al. [6]. This means one can understand both how a subject-level omics profile changes with time as well as understand differences in disease states within the same experiment. Further, extending experiments beyond two consecutive collection times can yield insights into whether and how omics profiles evolve with time. Herein, we propose a framework to model the evolution of a multivariate system for studies with two or more repeated omics measures. We define a multivariate system as one where multiple measures (e.g., gene expressions of multiple genes or microbial abundances of several bacteria) are collected from each experimental sample and where understanding the simultaneous changes of multiple variables is considered [7]. To demonstrate the utility of our model, we focus on microbiome data because of the inherent community dynamics that have a known tendency to respond to disease states [8,9,10].

Many studies have sought to measure the overall stability of the microbiome using a cross-sectional study design [11,12,13,14,15]. However, cross-sectional studies have their limitations, including a lack of ability to account for the noise introduced by microbiomes changing over time. Although there is a growing number of publications that include longitudinal collection times, the methods by which one can assess community stability are limited and inconsistent [6,16,17,18,19,20]. Typically, univariate changes in single species or genera are analyzed across time; however, these types of analysis do not answer how individual taxa contribute to the overall community dynamics due to a lack of accounting for correlated responses. Another longitudinal approach is *OmicsLonDA,* a non-parametric approach developed by Metwally et al. for omics studies [17]. This method considers longitudinal omics data and identifies the time intervals where omics profiles vary between study groups using a univariate approach [17]. Our model takes a different principled approach by combining multivariate, distance-based analysis with stochastic process modeling of longitudinal changes in high-dimensional datasets.

Distance-based approaches have shown utility in omics and multi-dimensional data analysis. In microbiome data, these methods allow for quantification of the similarity and dissimilarity between microbial communities and comparisons across different conditions, environments, and treatments. One of the most common distance-based methods is principal coordinate analysis (PCoA). PCoA is a dimensionality-reduction technique that projects high-dimensional arbitrary distance matrix onto a Euclidean space and allows for a lower-dimensional representation of the data along coordinates, maximizing the variance explained by the first few coordinates [21,22]. PCoA preserves the original pairwise distances between the samples and gives a representation of the data in a Euclidean space. This allows for the visualization of the relationships between samples in a more intuitive way and the identification of groups of samples (or different conditions) that are similar to each other. PCoA plots are often accompanied by statistical test results that quantify and assess the statistical significance of group differences in distance matrices, including the $T_{W}^{2}$ and $W_{d}^{*}$ tests [23,24]. The $T_{W}^{2}$ test offers a multivariate distance-based implementation of the Welch *t*-test of two means with unequal variance, and thus is suitable for two-group comparisons [23]. $T_{W}^{2}$ was developed to overcome challenges with alternative testing procedures, which were shown to have inflated type I error rates under heteroscedastic and unbalanced conditions. When used with Euclidean distances, $T_{W}^{2}$ provides a permutation-based unequal variance Welch equivalent of the Hotelling’s T-squared test. The $W_{d}^{*}$ test offers an extension to compare more than two groups simultaneously [24,25]. Similarly, when Euclidean distances are used, $W_{d}^{*}$ is a permutation-based Welch’s MANOVA test. Note that although PCoA is commonly used as a visualization for $T_{W}^{2}$ or $W_{d}^{*}$ analyses, it is not required for these tests.

Methods like $T_{W}^{2}$ and $W_{d}^{*}$ allow for quasi-longitudinal designs by allowing for repeated-measures (e.g., pre vs. post) comparisons using restricted permutations to determine significance values. However, this approach is limited because it does not easily accommodate multiple time points or variability in the interpoint times across observations. Nonetheless, irregular time series commonly occur in practice due to variation in sample collection intervals across subjects. To address this, we propose a multivariate drift-diffusion framework, MD3F, that takes a distance-based approach to model multivariate data over arbitrary-length time intervals using a drift-diffusion stochastic process. We show that inference and estimation of the magnitude of drift can be obtained based on just pairwise distance summary statistics of the multivariate data. Surprisingly, inference and estimation can be accomplished within by fitting generalized linear models.

In this paper, we describe the derivation of key estimates under the MD3F, a simulation study to explore type 1 error and statistical power characteristics of our approach. We further apply our method to two different microbiome datasets. The first are gut samples collected from infants predisposed to Type 1 diabetes [26]. The second are oral swabs collected from head and neck cancer patients before and after their treatment. The results of these applications support that the methodology will help understand the multivariate trajectories for high-dimensional patient data and how this can vary between patient groups or on a subject level. Overall, we determined that MD3F is a robust method for high-dimensional longitudinal datasets, but further work for biologic applications should be better characterized in future studies.

## 2. Materials and Methods

### 2.1. Model Description

To describe longitudinal evolution of a multivariate microbiome composition (described using a β-diversity dissimilarity), we distinguish between two types of random walks. First is a pure diffusion random walk, which causes the composition to change position randomly over time. These random walks do not have a specific preferred direction of change. Second is a drift-diffusion random walk, which, in addition to the first type of change, involves is a systematic force to change the composition in a specific direction. Thus, the drift-diffusion random walk includes both a random and a systematic component. The systematic component of such random walks is called the drift, which is a vector quantity and is thus characterized by a direction and a magnitude. MD3F is concerned primarily with the magnitude of the drift. In practice, the systematic component may consist of the tendency for certain microbial abundances to increase in composition and some other microbial abundances to decrease in a time-dependent manner. The motivation for our framework can be drawn from the illustration in Figure 1, which shows hypothetical steps of analysis of trajectories that do or do not have a drift. Our statistical goal is to establish the presence of the drift and, if possible, to estimate its magnitude. In this process, we are not necessarily concerned with determining the direction of the drift. It is sufficient to just establish that it is present. In Figure 1, Panel A, two trajectories are demonstrated, one with and one without drift. If we take a naïve approach to our problem, we may fit linear models along each dimension (x and y) regressed onto time and determine whether there is a drift in each dimension based on a coefficient test. This approach, however, is somewhat problematic not only because it may be prone to false positives and needs to be properly adjusted for multiple comparisons but also because it is not unique. Our answer may be different based on which projections of the data we consider. Figure 1, Panel B, shows a different projection of the same random walks in an orthonormal basis that puts the true drift (according to our simulation parameters) along the x axis. The MD3F approach is to calculate summary statistics-based interpoint distances across time intervals and use those statistics for an omnibus inference on the presence of drift and an estimation of its magnitude (Figure 1C).

More formally, let $M_{s}$ track the values of measured features (microbiome composition, gene expression, etc.) at time s following a drift-diffusion process in p dimensions (Figure 1A). Then, the change in finite time t, ${X_{t}=M}_{s+t}-M_{s}$ is distributed as N(αtW,Ξ), where α≥0 is the magnitude of the drift or the speed with which the change is happening, the direction of the change is unit vector W such that $W^{T}W=1$, and Ξ is the variance–covariance matrix for the features. Without loss of generality, we conduct a change of basis transformation of $X_{t}$ such that the change vector is V=(1,0,…,0) (Figure 1B). We call this variable $Y_{t}\sim N(\alpha tV,\Sigma )$. This transformation only changes the axes along which we view the change but does not alter the distances between points of the random walk trajectory. Then we can write the Euclidean distance of the features between time t as a standard quadratic form, which represents it in terms of standard normal random variables U~N(0,I):$$\mathrm{dist}{\left(M_{s},M_{s+t}\right)}^{2}={\left(M_{s+t}-M_{s}\right)}^{T}\left(M_{s+t}-M_{s}\right)=X_{t}^{T}X_{t}=Y_{t}^{T}Y_{t}={\left(U+b\right)}^{T}\Lambda \left(U+b\right),$$
where the original variance covariance is decomposed via diagonalization $\Sigma =P^{T}\Lambda P$ such that $P^{T}P={PP}^{T}=I$; $\Lambda =\mathrm{diag}\left(\lambda_{1},\ldots ,\lambda_{p}\right)$, where $\lambda_{i}$ are eigenvalues and P is the matrix of eigenvectors; and $b=\alpha tP\Sigma^{-1/2}V\Sigma =P^{T}\Lambda P$. Let us consider the expansions of the terms of the equation ${\left(U+b\right)}^{T}\Lambda \left(U+b\right)=b^{T}\Lambda b+U^{T}\Lambda U+U^{T}\Lambda b+b^{T}\Lambda U:$
$b^{T}\Lambda b={\left(\alpha tP\Sigma^{-\frac{1}{2}}V\right)}^{T}\Lambda \alpha tP\Sigma^{-\frac{1}{2}}V=\alpha^{2}t^{2}V^{T}\Sigma^{-\frac{1}{2}}P^{T}\Lambda P\Sigma^{-\frac{1}{2}}V=\alpha^{2}t^{2}V^{T}\Sigma^{-\frac{1}{2}}\Sigma^{\frac{1}{2}}{\Sigma^{\frac{1}{2}}\Sigma }^{-\frac{1}{2}}V=\alpha^{2}t^{2}V^{T}V=\alpha^{2}t^{2}.$$U^{T}\Lambda U=\sum_{j=1}^{p}\lambda_{j}U_{j}^{2}=\sum_{j=1}^{p}G(\frac{1}{2},2\lambda_{j})\approx \sum_{j=1}^{p}N\left(\lambda_{j},2\lambda_{j}^{2}\right)=\sum_{j=1}^{p}\lambda_{j}+N\left(0,\sum_{j=1}^{p}\lambda_{j}\right)$, where $G\left(\frac{1}{2},2\lambda_{j}\right)$ is a Gamma random variable. Here, the approximation is based on convergence of the Gamma distribution to the Gaussian distribution for large $\lambda_{j}$. The consequence of this approximation is that we can write this term as a scalar equation $U^{T}\Lambda U\approx r+N\left(0,x\right)$, consisting of an intercept r and a normal error, neither of which is dependent on α.Finally, note that $U^{T}\Lambda b$ and $b^{T}\Lambda U$ are linear combinations of uncorrelated Gaussian distributions with a mean of 0, e.g.,$U^{T}\Lambda b={\alpha tU}^{T}\Lambda P\Sigma^{-1/2}V=\alpha t{\left(\begin{array}{c} \lambda_{1}U_{1}\\ \lambda_{2}U_{2}\\ \begin{array}{c} \vdots \\ \lambda_{p}U_{p} \end{array} \end{array}\right)}^{T}P\Sigma^{-\frac{1}{2}}\left(\begin{array}{c} 1\\ 0\\ \begin{array}{c} \vdots \\ 0 \end{array} \end{array}\right)=\alpha t\sum_{j=1}^{p}z_{j}U_{j}=N\left(0,\alpha^{2}t^{2}\sum_{j=1}^{p}z_{j}\right)=N\left(0,z\right),$ where z is the variance.

Putting these derivations together, we can write:
(1)$$dist(M_{s},M_{s+t})=r+\beta t^{2}+e,\;\mathrm{where}\;e\sim N(0,\phi^{2}),\;\mathrm{and}\;{\beta =\alpha }^{2}$$

For clarity, $\phi^{2}$ is the variance of regression errors *e* and *r* is the intercept that summarizes the terms unrelated to time.

Note that because we can find a Euclidean representation of any distances using PCoA, the result above is applicable to drift diffusion with arbitrary distances [25]. Further, any dissimilarity can be used in place of a distance metric [27].

### 2.2. Inference and Estimation under the MD3F

The subsection above shows that the relationship between squares of interpoint distances and squares of time is linear. The corresponding linear estimator can be obtained from ordinary least squares regression. We call it *MD3Flinear2*. It is interesting to evaluate whether taking the squares is necessary; therefore, we also considered an estimator that does not square either: *MD3Flinear*.

### 2.3. Alternative Inference Approaches

It was pertinent to test comparable methods against MD3F; therefore, we compared two versions of the MD3F estimators (described in Section 2.2) to other bottom-up univariate approaches. To do so, we considered several alternatives:

*PCo1*: The PCoA-based Euclidean representation of the data was obtained and the first principal coordinate (PCo) was regressed on time to determine whether there was a time dependence in the observations using the coefficient *t*-test.

*PCo2*: Similar to above, the significance values for the first two PCos were obtained and the smallest of the two was used for overall significance.

*PCoA*: Similarly, all the minimums of all significance values for PCoA were used.

*uFDR*: Each variable/feature was regressed on time and the significance values were calculated using the coefficient *t*-test. The minimum *p*-value adjusted for multiple comparison using the false discovery rate procedure was used as the overall significance values.

*uHolm*: Similar to above, the univariate *p*-values were obtained and adjusted for multiple comparisons using Holm’s method.

### 2.4. Simulation Study

For simplicity and without loss of generality, we simulated multivariate data under the pre vs. post longitudinal model with variable time between observations. We assumed a baseline (pre) time of 0 and started by drawing a random sample of observation times s for the post timepoint for each sample. Here, *s* is drawn from an exponential distribution to allow for variability of interpoint times. Since ultimately only the distances between observations matter without loss of generality, we assumed that the pre observation was at the multivariate origin. Next, we generated the observations at the post by drawing a *d*-dimensional uncorrelated multivariate normal sample with means of $\alpha \times s/\sqrt{d}$ and the covariance matrix Σ=s×I. Under Equation (1), this yielded a realization of the drift-diffusion process over time interval *s*. Note that this choice ensures that the square of the mean interpoint Euclidean distance is $\sum_{i=1}^{d}{\left(\alpha \times \frac{s}{\sqrt{d}}\right)}^{2}=d\times \frac{\alpha^{2}s^{2}}{d}={\left(\alpha s\right)}^{2}$. Under this scenario, without loss of generality, the drift vector is $V=1_{d}/\sqrt{d}=(\frac{1}{\sqrt{d}},\ldots ,\frac{1}{\sqrt{d}})$, which means that the change associated with the process is propagating in equal magnitude along every dimension of the space. This description is general up to the change of basis.

Given that we focused the application of our method to microbiome datasets, we did not simulate our data with any missing values. The 16s rRNA gene amplicon sequencing-based microbiome assays, although typically undersampled, generally do not contain missing values in a strict sense. Further, if a dataset indeed has missing values, this may restrict one’s ability to calculate interpoint dissimilarities. Nonetheless, R statistical language has methods for handling missing values in dissimilarity calculations. An alternative approach may involve imputation; however, such methods have not been properly evaluated in the context of distance-based multivariate analyses.

We conducted the simulations controlling for (i) *the number of dimensions* d (1, 10, 100, and 1000), (ii) *sample size* n (10, 20, 50, 100), and (iii) *drift magnitude*
α (0, 0.5, 1, 2, and 3). We replicated the simulation 10,000 times for each set of parameters. Each replicate included a random walk trajectory that was analyzed simultaneously to infer the presence and estimate the magnitude of the drift.

We used the fraction of rejected null hypotheses to evaluate the type I error (when α=0) and power. For estimation, we further evaluated the bias as (MD3F Coefficient—Actual Speed) for MD3Flinear and (MD3F Coefficient—Square Actual Speed) for MD3Flinear2 models.

### 2.5. Reference Implementation and Code Availability

For our simulation study and all applications, we utilized R-studio (R version 4.1.3 and RStudio version 1.3.1093) [28]. Microbial data were organized and analyzed using the R-package phyloseq (version 1.38.0). Plots were generated in R-studio (version 1.3.1093, R version 4.1.3) with the ggplot2 package (version 3.3.5). Principal coordinate analysis was conducted using R package ade4 (version 1.7.18). A coding sample for the processing of this dataset and our application are both made available at (see Data Availability Statement).

### 2.6. Applications

#### 2.6.1. Microbiome Changes in Diabetes Onset

Infant gut microbiome sequencing results were made publicly available in Kostic et al. (https://doi.org/10.1016/j.chom.2015.01.001 (accessed on 29 March 2023)) [26]. The downloaded data were loaded into phyloseq without further preprocessing. We first considered the microbial sample sums for each individual sample and noticed that a portion of the samples had a total of less than 1. We decided to filter samples at a value of 0.8, dropping from a total of 777 to 672 samples in 33 subjects. We performed a center-log transformation on the OTU table before calculating the distance matrix. No other preprocessing was performed on these data. The sample data included with the Kostic et al. data provided the needed sample collection times. These data were modeled using MD3Flinear2 estimation. A coding sample is provided; see the Data Availability Statement.

#### 2.6.2. Microbiome Changes in Oral Mucositis Onset and Severity

Oral microbiome swabs were provided by our collaborators. Samples were collected in relation to their HNC treatment and the development of oral mucositis, a side effect of treatment. The first sample was collected just prior to the start of treatment, and the second at the onset of oral mucositis. All patient identifiers, including age, race, and sex, were eliminated. Oral microbiome samples were collected bilaterally from the buccal mucosa. Genus-level taxonomy was utilized for distance measures.

## 3. Results

### 3.1. Empirical Evaluation of Statistical Properties of MD3F

We first evaluated the validity of our method by estimating the empirical type I error rates with data simulated under the null hypothesis of α=0. A drift of zero represents a pure diffusion process, meaning there is no systematic direction to evolution of the data over time. In Figure 2A, we show the null hypothesis rejection rates at significance level of 0.05 of all tested methods. A valid method is expected to result in rejection frequencies at this nominal level. We observed that MD3F, both the squared and the non-squared version, were able to control for type 1 error across the range of dimensions and sample sizes. This is in contrast to the other methods tested, which failed to produce the expected rejected rates. We then further evaluated the MD3F methods by comparing performance across the different dimensions simulated. No clear patterns emerged based on the number of dimensions (Figure 2B), suggesting that these methods were valid across all parameters tested.

### 3.2. Evaluation of Model Power & Bias

Next, we evaluated the power of the MD3F approaches. To test the power to detect our alternative hypothesis, we simulated effect sizes (drift) of 0, 0.5, 1, 2, and 3. Non-zero effect size indicates there is a drift, and therefore we would expect to reject the null at the above-nominal significance level, which is what we see across all effect sizes in Figure 3A,B. In both versions of the model, increasing the dimensions of the drift resulted in lower power relative to smaller dimensional examples with the same number of observations due to each dimension introducing additional noise. Likewise, the increasing magnitude of the drift made it more detectable and resulted in higher estimated power in simulations. Overall, the method appears to have power in realistic sample size and dimensionality scenarios given an effect of appropriate magnitude was present.

Estimating the bias helped us to understand where performance faltered for the MD3F model. We report the average bias across simulations. Unlike in our power analysis, we observed a difference between the two MD3F methods when evaluating bias.

We observed that the MD3Flinear was more biased than the squared version, MD3Flinear2. In the MD3FLinear case, we saw an increasing amount of bias with both the number of dimensions and increased drift, whereas the squared version of the model seemed to act consistently at the different effect sizes (Figure 3). In both cases, increasing the sample size resulted in a reduction in model bias variance. Overall, the MD3Flinear2 approach appeared to be unbiased; therefore, we proceeded forward with this version of our model for biological applications.

### 3.3. Demonstration of MD3F through Application to Microbiome Datasets

We first applied our model to data publicly available from the Kostic et al. study conducted in 2015 [26]. In this study, stool samples were collected for sequencing periodically for 33 infants beginning at their birth. The purpose of this study was to determine whether the gut microbiome had any relationship to the onset of type 1 diabetes. This offered an ideal dataset to apply to our model, with there being a relatively high sample size for each individual. Additionally, it allowed for us to consider the study question in a novel way: How does each subject’s microbial composition change during early development? Such insights can help answer whether there is a meaningful change in the gut microbiome prior to type 1 diabetes onset. We plotted the estimated drifts for four individuals by diabetes status, as seen in Figure 4A (all individual-level trajectories are provided in Figure A1). In Figure 4B, we summarize the estimated drifts for all subjects by case status. Although no significant difference (student’s *t*-test, *p*-value = 0.49) was observed in the estimated drifts between groups, it is worth noting that there was an observed drift in most subjects. The estimate of the average drift in controls was smaller than in cases, at 1.1597 and 1.1591, respectively. Determining the drift across subjects helps to understand whether there is a time-dependent change in the gut microbiome, meaning that we can understand whether there is a natural evolution of the gut microbial community or one specifically associated with a disease state.

Our second application was to oral microbiome swabs collected from head and neck cancer patients before their cancer treatment beginning and at the onset of oral mucositis. This study had special interest in oral mucositis, as it is a detrimental side effect of chemotherapy and radiation treatment. Rather than applying our method to all subjects in this case, since we were focused on only two time points, we applied our model across all subjects but found no detectible drift (Figure 5). However, we did observe an interesting pattern where the patients with the earliest time to onset of oral mucositis (i.e., 14 days or less) had an overall larger interpoint distance between the two timepoints. In other words, there was a larger change in the microbiome in a shorter period of time.

## 4. Discussion

We have developed a multivariate distance-based drift diffusion framework to model multivariate changes in omics profiles over time. MD3F is based on a novel Gaussian approximation of the relationship between the squares of the interpoint distances of the omics profiles over time. Distance-based approaches are capable of handling the complexity of multivariate spaces and trajectories, which was a motivation for our approach [29]. We evaluated this approximation and found our model to control type 1 error and to be powerful at detecting when effects are present in simulated data. The MD3F resulted in unbiased drift estimates.

We were also able to apply our method to biological samples collected from patients. Our model was applied to two different datasets, both of which were microbiome samples collected from patients over time. The application of MD3F to infant gut microbiome data helped determine that a time-dependent change in the microbial profile was not associated with type 1 diabetes status. Despite this, we still observed that most individuals had a detectible drift, but further analysis will help elucidate the significance of this finding. Our results are further validated by the author’s initial observation that α-diversity increased with time but did not vary between controls and cases for the majority of the study time [26]. In the application of MD3F to head and neck cancer patients, we did not detect a measurable drift. However, we did observe an interesting relationship in our data, which was that those with the shortest time to onset of oral mucositis had the greatest change in distance between timepoints. This finding supports that onset time of oral mucositis is non-random: An earlier, larger change leads to a shorter time to the development of a disease state. It is well characterized that the oral microbiome changes with head and neck cancer treatment in general but also with the development of oral mucositis [30,31]. Thus, seeing an earlier change can intuitively be associated with an earlier development of oral mucositis, supporting the basis of our methodology in this space. Although MD3F is concerned with estimating the magnitude of the drift itself, understanding the direction of the drift requires additional analyses. Some guidance for interpretability can be gleaned from Felli et al., where differentiation trajectories were studied for hematopoietic stem cells using transcriptomic data [32]. The authors use the Pearson correlation between CD34+ progenitor cells at differentiation time points, so a stronger negative correlation is associated with a more rapid differentiation [32]. It could be worth exploring how the change in the multivariate space relates to specific bacteria that are indicative of a healthy or diseased state.

It should be noted that although we chose to employ Euclidean and Aitchison (center-log transformation used with Euclidean distances) distances in our approach, other β -diversity and dissimilarities can be used with MD3F instead. Further, we utilized PCoA for visualization, but other projection techniques could be useful in this space and are worth exploring. Finally, although there is a magnitude of different approaches for lower dimensional projections, MD3F does not work in lower dimensional projections.

We believe our model has robust potential for comparable studies. This is because (1) there is no currently available method to our knowledge to assess individual, multivariate trajectories on a patient level; (2) we do not rely on unsupervised methods for assigning patients to clusters, as in [33,34]; and (3) there is flexibility to compare aggregate patient samples or individual trajectories. Despite the need for greater characterization in biological data, we have shown that MD3F is robust to type I and type II error and that we can leverage a distance-based approach for longitudinal data.

## Figures and Tables

**Figure 1 genes-15-00582-f001:**
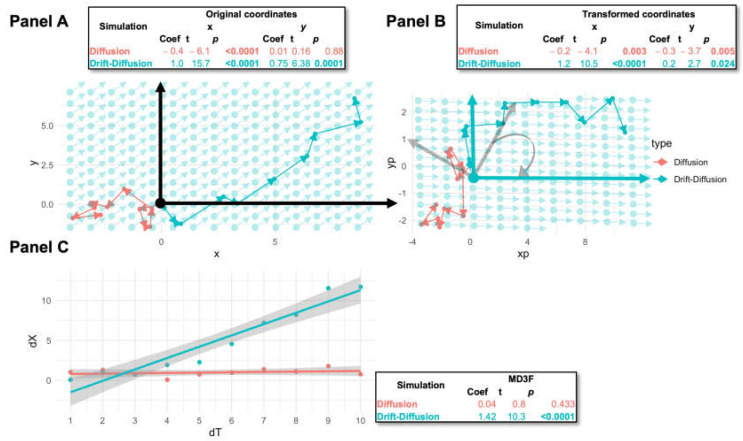
**MD3F analysis of random walks.** Panel (**A**) shows samples of pure diffusion (coral) and drift-diffusion (teal) random walks both initiating at <0, 0> and evolving on a unit time grid. The magnitude of the drift is set to 1 and it is pointed in the <1, 1> direction, indicated by faint teal lines. In panel (**B**), the same random walks are represented in a different orthonormal basis, where the first basis vector is in the direction of the drift (i.e., <1, 1>). The original basis is indicated by gray vectors. For both panel A and panel B, there are 10 time points for each random walk. Panel (**C**) shows the relationship between time-scaled squared interpoint Euclidean distances and the length of the time interval from the start of the random walk. These are the input data to the MD3F. The results of MD3F and univariate ordinary least-squares regression-based analyses of each of the random walks are shown in the insets.

**Figure 2 genes-15-00582-f002:**
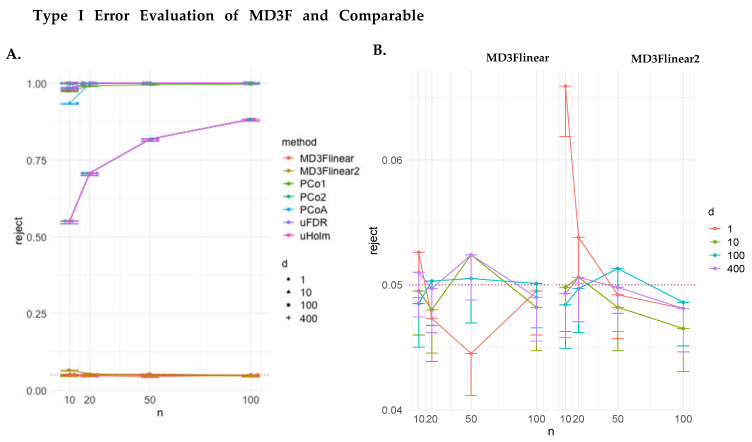
**Type I error evaluation of MD3F and comparable methods.** (**A**) Empirical type I error of MD3F estimators against other methods. MD3Flinear fits the d∝αt linear regression model, whereas MD3Flinear2 fits $d^{2}\propto \alpha^{2}t^{2}$. The methods are evaluated over 1, 10, 100, and 400-dimensional data and sample sizes of n = 10, 20, 50, 100. The horizontal red dashed line at y = 0.05 is the acceptable threshold where we would expect models to perform. The MD3F methods are the only two that could control for type I error. (**B**) Empirical type I error of MD3Flinear and MD3Flinear2 estimators. Overall, the empirical statistical performance of both MD3F methods appears to be similar. The number of dimensions did not impact the performance for either method, except in univariate data, where MD3Flinear2 may have slight type I error inflation in small samples.

**Figure 3 genes-15-00582-f003:**
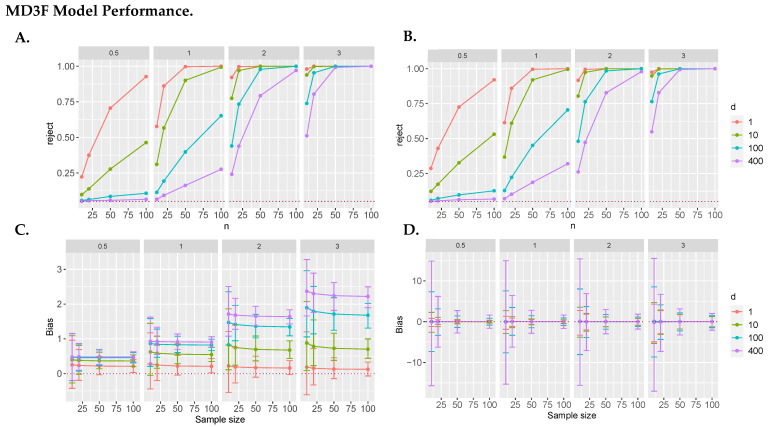
**MD3F model performance.** (**A**,**B**) Power to detect drift using MD3F. (**A**) MD3FLinear results are reported. (**B**) MD3FLinear2 results are reported. The dotted line indicates nominal significance level. Each panel represents the effect size or speed of the drift. We see that both models have adequate power to reject the null. (**C**,**D**) MD3 Model bias assessment. (**C**) MD3FLinear results reported. (**D**) MD3FLinear2 results reported. The dotted line is a y = 0, indicative of no bias. MD3FLinear2 shows no evidence of bias. There also appears to be a relationship in MD3FLinear where increasing the number of dimensions increases the bias.

**Figure 4 genes-15-00582-f004:**
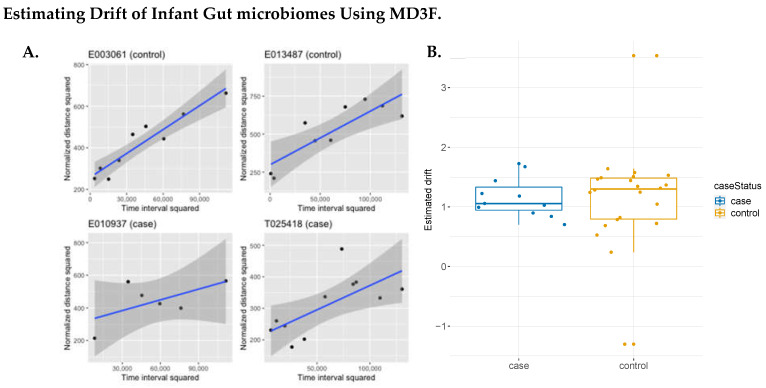
**Estimating the drift of infant gut microbiomes using MD3F.** (**A**) Normalized distances measured from four infant microbiome samples over time. The fitted model for each subject is indicated with a blue line. The shadows represent the 95% confidence interval for each model. All four subjects show a positive drift over time regardless of diabetes status. (**B**) MD3 predicted drift estimates by diabetes status for all subjects. The estimated drift for each subject was recorded and then compared to diabetes status. We do not observe a difference in the drift rates between groups.

**Figure 5 genes-15-00582-f005:**
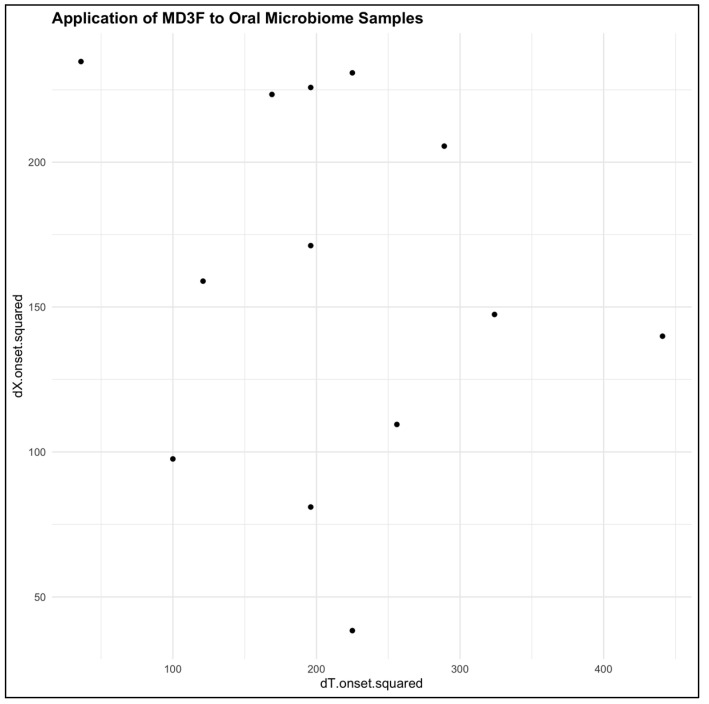
**Application of MD3F to oral microbiome samples.** Each dot represents an individual subject. The population drift rate from baseline to the onset of oral mucositis was not significant (estimate = 2.55, *p*-value = 0.79). Despite there being an obvious linear trend, it is notable that with an earlier time to onset of oral mucositis (i.e., 14 days or less), there is generally a large change in the distance moved from baseline.

## Data Availability

The data presented in this study from Kostic et al. are available at https://doi.org/10.1016/j.chom.2015.01.001 (accessed on 29 March 2023). Coding examples for simulation and application studies are made available online at https://doi.org/10.5281/zenodo.11040585 (accessed on 25 April 2024).

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
