# Peer review of "MD3F: Multivariate Distance Drift Diffusion Framework for High-Dimensional Datasets"

_genes, 2024, doi:10.3390/genes15050582_

Round 1
Reviewer 1 Report
Comments and Suggestions for Authors The manuscript describes a potentially impactful new method for analyzing longitudinal high-feature datasets with a demonstration on a microbiome dataset and rigorous evaluation through simulations. The method itself is well demonstrated; however, there are major issues that need to be addressed in this manuscript.Major Issues 1) The authors should add their data, code, and full results as supplemental material to fully support scientific reproducibility. A GitHub repo can be altered or deleted at any time and thus does not provide the permanence needed to support scientific reproducibility. Figshare and Zenodo items provide a good mechanism to provide complex supplemental material with a DOI. The authors should review relevant literature on best computational practices to support scientific reproducibility. For example: Stodden V, McNutt M, Bailey DH, Deelman E, Gil Y, Hanson B, Heroux MA, Ioannidis JP, Taufer M. Enhancing reproducibility for computational methods. Science. 2016 Dec 9;354(6317):1240-1.
2) Line 78: Authors need to add more description of the T^2_w and W^*_d tests and their relationship to classical statistical test and methods. For example, the T^2_w test could be called a Welch equivalent to the Hotelling’s T-squared test and the W^*_d test is known as the Welch’s MANOVA test. Would be good to indicate that the T^2_w test is an unequal variance Welsh equivalent of the Hotelling’s T-squared test. Also, the authors should explain how the PCoA helps to satisfy the independence requirements of the T^2_w and W^*_d tests. Also, the authors should do due diligence to get original references to the given statistical test instead of citing just a paper that applies the statistical test. For example, the proper reference for the T^2_w test is: Alekseyenko, A.V.: Multivariate welch t-test on distances. Bioinformatics 32(23), 3552–3558 (2016). doi:10.1093/bioinformatics/btw524 Given that the corresponding author is the author of this reference, it is probably an oversight.
3) Line 108: Should better describe what the concept “drift” means and represents in this context.
4) The journal Genes is not a statistics journal. The authors should define ALL of their variables in there equations to improve readability for a broader audience. For example, alpha, phi, e, lambda, z, etc. are not defined, making these equations much harder for non-statisticians to understand.
5) There is no description of how missing values were handled in the microbiome datasets. Were they dropped from the analysis? Were they imputed? How were they handled? What fraction of the dataset was missing? Missing values can have a significant effect on data analysis. This reviewer assumes the simulated datasets did not include missing values.
6) It is unclear how many longitudinal time points were used in the simulations. Figure 1C implies 10 time points, but this is never explicated written. This reviewer assumes that sample size n refers to the number of biological replicates. It seems that the effect of the number of time points used in the simulations should be adequately explored too.
Minor Issues: Line 118: add a comma after “our problem”. Line 275: add a comma after “In Figure 4B”. Line 277: change “different” to “difference. Also indicate whether the sentence refers to statistical significance or effect size, since the term “significant difference” is very ambiguous. Comments on the Quality of English Language
See above.
Reviewer 2 Report
Comments and Suggestions for Authors
Distance based approaches are very powerful methods when dealing with multivariate spaces. These methods are intrinsically non-linear (by the way whatsoever distance has a circular simmetry as evident by the same definitioin of a circle as a set of points with distance = r to a reference point we define as 'center'). This allows to focus on both 'isometric' (all the pairwise distances between the elements are linearly correlated across time) trajectories and 'allometric' (or shape changing) evolution in time (lack of correlation among pairwise distance). This was a basic tenet of distance geometry as applied to development trajectories since many decades and allows to face highly non-linear phenomena in a straightforward way (see Székely, Gábor J., Maria L. Rizzo, and Nail K. Bakirov. "Measuring and testing dependence by correlation of distances." (2007): 2769-2794.).
The authors dare to revive a classical way of thinking the majority of biologists forgot and provide a very rigorous mathematical frame for inferential analysis of distance based longitudinal data. The only missing point, in my opinion, is that they do not take explictly into account the 'shape changing' and 'shape invariant' aspects of longitudinal distance data. This can be done by going in depth into pairwise between samples distances (see https://link.springer.com/article/10.1186/1752-0509-4-85). In the mentioned paper a isometric (keeping invariant the relative positions of the samples in the multidimensional scaling derived space) trajectory coming from transcriptome data is compared with an allometric countepart relative to miRNA. I think the manuscript could benefit from such a reference to shape invariant and shape preserving trajectories (that is crucial for distance geometry approaches in structural biochemistry).
Another point to be addressed is the one stressed in the below attached Sneath and Sokal paper and has to do with the robustnesss of distance spaces with respect to missing data and change of reference points. This can be crucial in many biological applications. Minor points are related to the fact that PCoA (or metric multidimensional scaling) is not the only technique to project a distance space into a low dimension explicit space, as a matter of fact Bray-Curtis ordination and non-metric multidimensional scaling are much more common in ecology and microbiome studies, but this is a minor aspect.

Round 2
Reviewer 1 Report
Comments and Suggestions for Authors
The authors have adequately addressed my concerns.